

# Expression of neutrophil extracellular trap-related proteins and its correlation with IL-17 and TNF-α in patients with oral lichen planus

Juehua Cheng[1,2,3,*], Chenyu Zhou[2,3,4,*], Jia Liu[1,2,3], Yanlin Geng[1,2,3], Lin Liu[1,2,3] and Yuan Fan[1,2,3]

[1] Department of Oral Mucosal Diseases, The Affiliated Stomatological Hospital of Nanjing Medical University, Nanjing, China
[2] Jiangsu Province Key Laboratory of Oral Diseases, Nanjing Medical University, Nanjing, China
[3] Jiangsu Province Engineering Research Center of Stomatological Translational Medicine, Nanjing, China
[4] Qingpu Branch of Zhongshan Hospital, Fudan University, Shanghai, China
* These authors contributed equally to this work.

Corresponding authors
Lin Liu, 15250969013@163.com
Yuan Fan, fanyuan@njmu.edu.cn

## ABSTRACT

**Background:** Neutrophil extracellular traps (NETs) are produced by polymorphonuclear neutrophils (PMNs) stimulated by interleukin-17 (IL-17) and tumor necrosis factor α (TNF-α). However, the level and role of NETs in oral lichen planus (OLP) remain poorly understood.

**Objective:** This study aimed to investigate the expression of NETs in OLP and explore the correlation between NETs and the levels of IL-17 and TNF-α.

**Methods:** The expression and distribution of NET-related proteins in tissue samples from each group were assessed using hematoxylin-eosin (HE) staining and immunofluorescence (IF). Additionally, the expression of NET-related proteins in peripheral blood samples from each group was evaluated using cell IF technique and fluorescence spectrophotometry. The relative formation level of NETs in each group was determined by fluorescence spectrophotometry *via* plasma co-culture. Furthermore, the levels of inflammatory cytokines IL-17 and TNF-α in plasma and culture supernatant were measured using enzyme-linked immunosorbent assay (ELISA).

**Results:** NET-related proteins were located in the subepithelial and lamina propria layers of OLP lesions. OLP had significantly higher expression of NET-related proteins in lesion tissues and peripheral blood compared to the healthy control (HC) group ($p < 0.05$). The rate of NETs formation in the erosive-stage OLP (EOLP) group was significantly higher than that in the HC group ($p < 0.05$), in contrast, no significant increase was observed in the non-erosive OLP (NEOLP) group ($p > 0.05$). Furthermore, the levels of IL-17 and TNF-α in the EOLP group were significantly elevated compared to those in the NEOLP group and HC group ($p < 0.05$), while the levels in the NEOLP group did not significantly differ from those in the HC group ($p > 0.05$). The rate of NETs formation showed a positive correlation with the levels of IL-17 and TNF-α in plasma.

**Conclusion:** The expression of NET-related proteins was upregulated in OLP lesion tissues and peripheral blood. Elevated levels of IL-17 and TNF-α in peripheral blood

plasma positively correlated with the rate of NETs formation, suggesting that IL-17 and TNF-α mediate the formation of NETs in OLP patients, and may thereby contribute to the development of OLP.

## INTRODUCTION

Oral lichen planus (OLP) is a common chronic inflammatory disease affecting the oral mucosa, characterized by localized autoimmune reactions, with a global prevalence estimated at 1.01% (*González-Moles et al., 2021*). With a cancer rate of 1–2%, OLP has been classified as an oral potential malignant disorder (OPMD) by the World Health Organization (*Warnakulasuriya et al., 2021*). Clinically, OLP is usually divided into non-erosive oral lichen planus (NEOLP) and erosive oral lichen planus (EOLP) based on the presence of erosive lesions (*Chiang et al., 2018*). NEOLP typically presents with white or off-white reticular stripes known as Wickham's striae, while EOLP exhibits congestion and erosions in the surrounding mucosa. Previous research has underscored the pivotal role of immune factors, particularly local immune dysregulation, in the occurrence and development of OLP (*El-Howati et al., 2023*). There are a large number of inflammatory factors in OLP lesions, such as interleukins, TGF-β, interferon-γ (IFN-γ), TNF-α, and chemokines (*Lu et al., 2015*; *Rivera et al., 2020*; *Shan et al., 2019*, *2020*). In addition, various immune cells, endothelial cells, and fibroblasts within OLP tissues jointly constitute the pathological microenvironment and interact with inflammatory factors (*Costa et al., 2020*).

Polymorphonuclear neutrophils (PMNs) have a variety of biological functions, such as chemotaxis, phagocytosis, and immune regulation, which are related to the pathogenesis of oral inflammatory diseases (*Williams et al., 2021*). Neutrophil extracellular traps (NETs) trap and kill pathogenic microorganisms *via* electrostatic adsorption, serving as a crucial host defense mechanism (*Hidalgo et al., 2022*). The process of NETs formation in PMNs is called NETosis, representing a novel form of PMN death distinct from apoptosis and necrosis (*Brinkmann et al., 2004*). Myeloperoxidase (MPO), primarily localized within the azurophilic granules of neutrophils and abundant in the cytoplasm, exerts significant bactericidal effects, thus serving as a key marker of neutrophils (*Wang & Nauseef, 2022*). Circulating free DNA (cfDNA), an important component of NETs, triggers the body's immune response, and cfDNA/NETs are a novel marker for evaluating infection levels in the host (*Margraf et al., 2008*).

Recent studies have shown that in the early stages of periodontitis, neutrophils infiltrate the gingival mucosa, discharge NETs to trigger mucosal inflammation leading to the upregulation of interleukin-17 (IL-17)/Th17 response, consequently contributing to bone destruction (*Kim et al., 2023*). Furthermore, *Garley*'s *(2023)* review demonstrated that lipopolysaccharide (LPS) or IL-17 stimulation induces the formation of NETs in oral squamous cell carcinoma (OSCC), with elevated levels of NET markers detected in the

serum of patients with OSCC. In addition, elevated expression of NETs has been observed in various autoimmune diseases. Neutrophils from rheumatoid arthritis (RA) patients tend to produce NETs, with serum and synovial fluid serving as principal inducers of NET formation (*O'Neil & Kaplan, 2019*). In patients with systemic lupus erythematosus (SLE), peripheral blood neutrophils exhibit increased NETosis, with IL-17A and tissue factor (TF) modulating the composition of the produced NETs. NETs containing TF and IL-17A are also evident in cutaneous lesions and renal tissues of SLE patients (*Frangou et al., 2019*). Similarly, patients with psoriasis showed increased expression of NET-related proteins, coupled with a stronger trend for NETosis in peripheral neutrophils (*Herster et al., 2020*).

Currently, there have been few studies on NETs in OLP. Previous studies have only verified the phenomenon of increased NETs expression in the peripheral blood of OLP patients (*Jablonska et al., 2020*). As a localized autoimmune disease, it remains unclear whether there is a heightened expression of NET-related proteins in the local lesion tissue of OLP, necessitating further investigation. Additionally, OLP is also an inflammatory disease. Previous studies have shown that the expression of IL-17 and tumor necrosis factor-α (TNF-α) in damaged tissues and peripheral blood of OLP patients correlates with different clinical subtypes of the disease, indicating that the dysregulation of the local immune microenvironment plays a pivotal role in OLP pathogenesis (*Lu et al., 2015*). However, whether this dysregulation contributes to enhanced formation of NETs in OLP requires further validation.

Therefore, this study explored the role of NETs in the pathogenesis of OLP by investigating the expression of NET-related proteins in OLP tissues and peripheral blood, along with assessing the impact of inflammatory cytokines IL-17 and TNF-α in OLP plasma on the formation of NETs *in vitro*, aiming to provide novel insights into the clinical treatment of OLP.

## MATERIALS AND METHODS

### Patients and methods

Tissue and peripheral blood samples were obtained from individuals visiting the Department of Oral Mucosal Diseases at the Affiliated Stomatology Hospital of Nanjing Medical University, Jiangsu Province, between May 2021 and September 2022. According to the guidelines of the 5th edition of Oral Mucosa, edited by *Qianming (2020)*, and the recommendations proposed by *van der Meij & van der Waal (2003)*, a total of 20 patients with EOLP and 20 patients with NEOLP were recruited. Exclusion criteria: 1) having other established oral mucosal diseases; 2) having more severe systemic diseases, tumors; 3) having used antibiotics within 1 month and immunological agents within 3 months; 4) certain drugs or silver amalgam fillings that may cause moss-like reactions; 5) severe tobacco and alcohol addiction. Healthy subjects were included as having no systemic disease, no oral mucosal disease, and no antibiotics or immunosuppressive agents within 3 months. Within the erosion group, there were eight males and 12 females, with an average age of (46.8 ± 11.0) years. The NEOLP group consisted of nine male and 11 females, with an average age of (44.3 ± 13.6) years. The healthy control (HC) group

($n$ = 20) were selected from patients undergoing treatment in the implant department during the same period, including 10 males and 10 females, with an average age of (39.7 ± 3.9) years. None of the subjects exhibited oral mucosal diseases or serious systemic illnesses, nor had they taken antibiotics or immunosuppressants within 3 months. There were no significant differences in age or gender distribution among the groups ($p > 0.05$). Informed consent was obtained in writing from all participants before sample collection. All procedures were approved by the Ethics Committee of the School of Stomatology, Nanjing Medical University, under approval number No. 281 (2019).

### Immunofluorescence analysis of tissues

After the initial routine hematoxylin-eosin (HE) staining of the first section from each group, the remaining sections underwent tissue immunofluorescence (IF) staining to assess the levels of NET expression. Formalin-fixed and paraffin-embedded tissue sections (4 µm) were placed on adhesive slides (CITOGLAS, Jiangsu, China) and subjected to deparaffinization at 65 °C for 1 h. Deparaffinization was achieved using the xylene method, followed by rehydration with ethanol. Subsequently, the tissue sections were washed twice with phosphate-buffered saline (PBS; Gibco, Waltham, MA, USA), and antigen retrieval was performed using sodium citrate buffer (0.01 M, Phygene, Fujian, China) heated for 20 min. To minimize natural fluorescence, the sections were treated with 3% hydrogen peroxide for 10 min at room temperature. After two PBS washes, the sections were blocked with goat serum for 45 min at 37 °C and then incubated overnight at 4 °C with anti-MPO antibody (1:100, ab208670, Abcam, Cambridge, MA, USA). Following removal of the primary antibody and two PBS washes, the sections were incubated with CoraLite488-labeled goat anti-mouse IgG (H+L) secondary antibody (Proteintech, Wuhan, China) for 45 min at 37 °C in the dark. After two additional PBS washes, nuclei were stained with DAPI (Beyotime, Jiangsu, China) for 2 min. Following two more PBS washes, the slides were sealed with an anti-fluorescence quenching sealer (Beyotime, Jiangsu, China) and examined under a laser confocal microscope (Carl Zeiss, Oberkochen, Baden-Württemberg, Germany) at 20× magnification. Excitation wavelengths for the blue and red channels were set to 405 and 561 nm, respectively. Images were captured randomly using ZEN software (Carl Zeiss, Oberkochen, Baden-Württemberg, Germany), and the expression levels of NET-related proteins in tissues were analyzed semi-quantitatively by using ImageJ software version 1.53 m (National Institutes of Health, Maryland, USA).

### Cell isolation, identification and IF analysis

Freshly drawn peripheral blood was collected and neutrophils were isolated within 2 h using the Human Neutrophil Isolation Solution Kit (TBDScience, Tianjin, China). A total of 6 mL of human neutrophil isolate was added to a 15 mL tube and peripheral blood was placed above the liquid level and centrifuged at 650 g for 30 min. The liquid in the tube was divided from top to bottom into plasma, monocytes, neutrophils and erythrocytes along with other cells. Approximately 1 mL of plasma was aspirated into lyophilization tubes (Thermo Fisher Scientific, Waltham, MA, USA) and stored in a refrigerator at −80 °C for subsequent experiments. The lower layer containing neutrophils was washed with HBSS

washing solution, centrifuged at 350 g for 10 min, and the supernatant was discarded. The remaining cells were lysed using erythrocyte lysis solution, centrifuged at 250 g for 8 min, and the supernatant was discarded. Cell components were resuspended in 1 mL of RPMI 1640 medium (Thermo Fisher Scientific, Waltham, MA, USA) containing 10% Australian fetal bovine serum (FBS, Gibco, Waltham, MA, USA) and penicillin-streptomycin double antibody (HYclone, Logan, UT, USA). Cell viability (>95%) and purity (>95%) were confirmed by flow cytometry after cell counting.

NET-related proteins were detected using cellular IF, and the expression levels of NETs were quantified. The cell concentration was adjusted to $5 \times 10^5$/mL. Polylysine-treated cell slides (BOSTER, Wuhan, China) were placed in 24-well plates (NEST, Wuxi, China), and each well was seeded with 1 mL of cell suspension. The plates were then incubated overnight at 37 °C with 5% $CO_2$. After removing the supernatant, the cells were washed three times with PBS and fixed by adding pre-warmed 4% paraformaldehyde (Biosharp, Beijing, China) at 37 °C. Goat serum was used for blocking. The primary antibody was applied and incubated overnight at 4 °C, followed by incubation with the secondary antibody at 37 °C for 45 min in the dark. Nuclei were stained with DAPI, and the slides were sealed with an anti-fluorescence quenching sealer, following the same procedure as the previous tissue IF staining. Subsequently, the images were examined under a laser confocal microscope at 20× magnification, with excitation wavelengths set to 405 nm for the blue channel and 561 nm for the red channel. The number of neutrophils and NET-related proteins was recorded separately using ZEN software to calculate the formation rate of NETs.

## Quantitative analysis of NETs in peripheral blood

The expression levels of NET-related proteins were quantified using a fluorophotometric assay. A 96-well plate (NEST) was utilized, with each well receiving 100 μL of cell suspension, and two replicate wells were set up. SYTOX$^{TM}$ Green Nucleic Acid Dye (Thermo Fisher Scientific, Waltham, MA, USA) was added to each well to achieve a final concentration of 0.4 μM. The plate was then incubated in the dark at 37 °C in a 5% $CO_2$ environment for 5 min. Subsequently, the optical density (OD) value was read at 560 nm using a microplate spectrophotometer (Molecular Devices, San Jose, CA, USA).

## Co-culture of plasma and neutrophils and detection of NETs formation

Neutrophils from the HC group were co-cultured with plasma from each respective group to evaluate the impact of different plasma inflammatory environments on NETosis. The experiment comprised four distinct groups: a blank group (neutrophil culture medium only), the HC group (plasma from the HC group), the NEOLP group (plasma from the NEOLP group), and the EOLP group (plasma from the EOLP group). Neutrophil suspensions from the HC group were aliquoted into EP tubes (Biosharp, Harjumaa, Estonia) at 1 mL per tube for a total of four tubes. These tubes were centrifuged at 1,000 rpm for 5 min, after which the supernatant was discarded. Subsequently, the corresponding culture medium and plasma were added to the tubes, and the tubes were then incubated overnight at 37 °C in a 5% $CO_2$ environment.

Both the culture medium and plasma were clear and free of turbidity, and the cells exhibited good condition under an inverted microscope without any signs of contamination. Following centrifugation at 1,000 rpm for 5 min, the supernatant was aspirated into EP tubes and stored in the freezer at −20 °C for subsequent experiments. The cellular fraction was resuspended in 1 mL of culture medium and dispensed into a 96-well plate at 200 µL per well, with two replicate wells for each condition. Subsequently, 20 µL of SYTOX$^{TM}$ Green fluorescent dye was added to each well, and the plate was incubated at 37 °C with 5% $CO_2$ for 5 min. The optical density (OD) value was then measured at 560 nm using a microplate spectrophotometer.

### Enzyme-linked immunosorbent assay

The levels of IL-17 and TNF-α in the supernatant of each cell group after co-culture were quantified using enzyme-linked immunosorbent assay (ELISA). The supernatants were removed from −20 °C storage to room temperature and subsequently centrifuged at 2,000 rpm for 20 min. A standard curve was prepared as per the kit instructions, and the human IL-17 ELISA kit and human TNF-α ELISA kit (Mlbio, Shanghai, China) were used according to the manufacturer's instructions. Two replicate wells were established for each sample, and the absorbance readings at 450 nm were recorded using a multifunctional enzyme marker.

### Statistical analyses

The IF staining and the cell experiments were conducted in 20 cases. Negative controls without primary antibody were included in each staining batch to assess the specificity and background of the fluorescent signal. Student's t-test was applied for comparisons between two groups, and ANOVA followed by Tukey's *post-hoc* test for multiple comparisons. ImageJ 1.53 m software was used for semi-quantitative analysis of the relative fluorescence intensity of NET-related proteins in each group, and GraphPad Prism version 9 (Boston, CA, USA) was used for statistical analysis and mapping. A *p*-value of <0.05 was considered statistically significant.

## RESULTS

### HE staining and IF analysis of NET-related protein in OLP lesions

MPO is located on the azurophilic granules of neutrophils, playing a pivotal role in histone cleavage and chromatin decondensation, thereby contributing to NET formation. This section adopted tissue IF technology to qualitatively assess the expression of NETs in tissues by examining the DNA-MPO complex expression under laser confocal microscope observation (*Byrd et al., 2019*).

The results from both HE staining and IF staining revealed an obvious increase in MPO-positive staining (depicted in red) within NEOLP and EOLP tissues, co-localizing with DNA structures (depicted in blue) to form MPO-DNA complexes, indicating the presence of NET-related proteins. In contrast, elevated levels of NET-related proteins were not observed in the normal control group. The white arrows show the distribution of

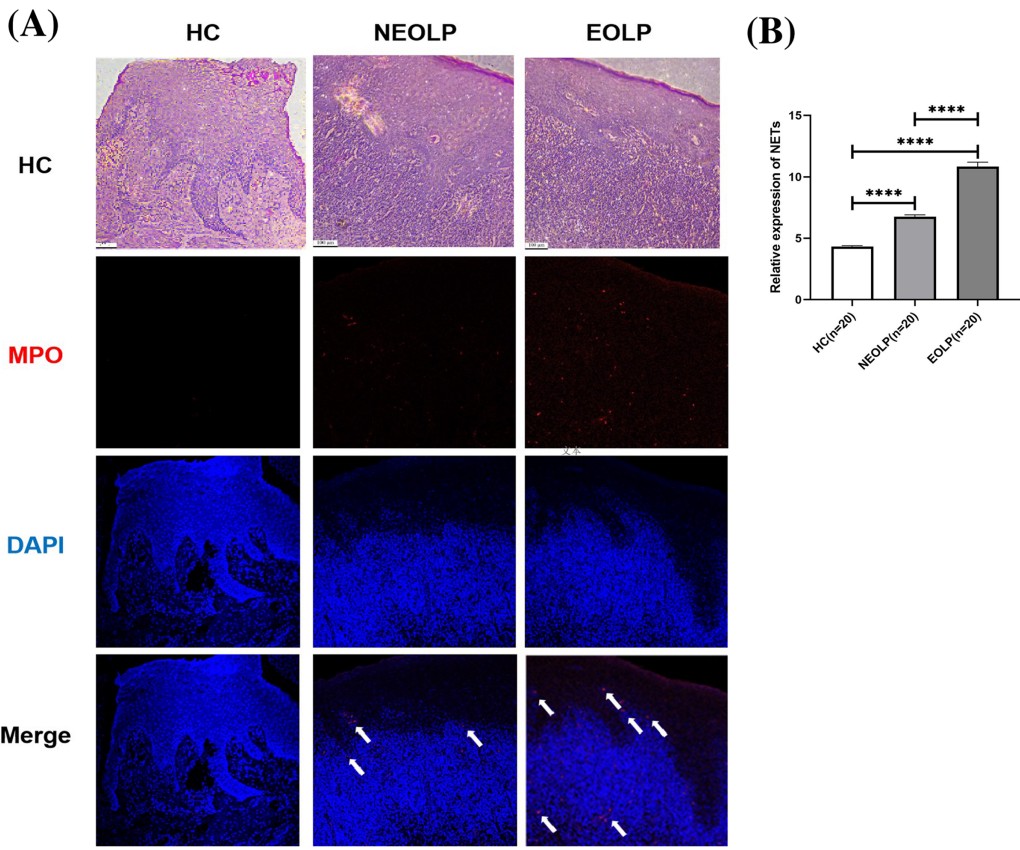

**Figure 1** (A) Hematoxylin and eosin staining and immunofluorescence staining showing MPO-DNA expression in tissue. Scale bar: 100 μm. (B) Expression of MPO-DNA, a NET-related protein, in tissue. MPO, myeloperoxidase. **** means $p < 0.0001$. 

NETs. Combined with HE staining, the increased expression of NET-related proteins mainly appeared in the subepithelial and lamina propria regions of the mucosa (Fig. 1A).

Through semi-quantitative analysis of the relative fluorescence intensity of red fluorescence in each group, it was found that the expression of NET-related proteins in the NEOLP and EOLP groups was significantly higher than that in the HC group ($p < 0.0001$). Moreover, within the EOLP group, the expression was significantly higher than that in the NEOLP group ($p < 0.0001$) (Fig. 1B).

## Qualitative and quantitative analysis of NET-related protein in peripheral blood

The results of cellular IF staining indicated a noticeable presence of MPO-DNA complexes in NEOLP and EOLP tissues, while their expression was not obvious in normal peripheral blood (Fig. 2A). The expression of NET-related proteins in the peripheral blood of NEOLP and EOLP patients was significantly higher than that in normal peripheral blood ($p < 0.05$). Interestingly, the expressions in the EOLP and NEOLP groups were similar, with no significant difference observed ($p > 0.05$) (Fig. 2B).

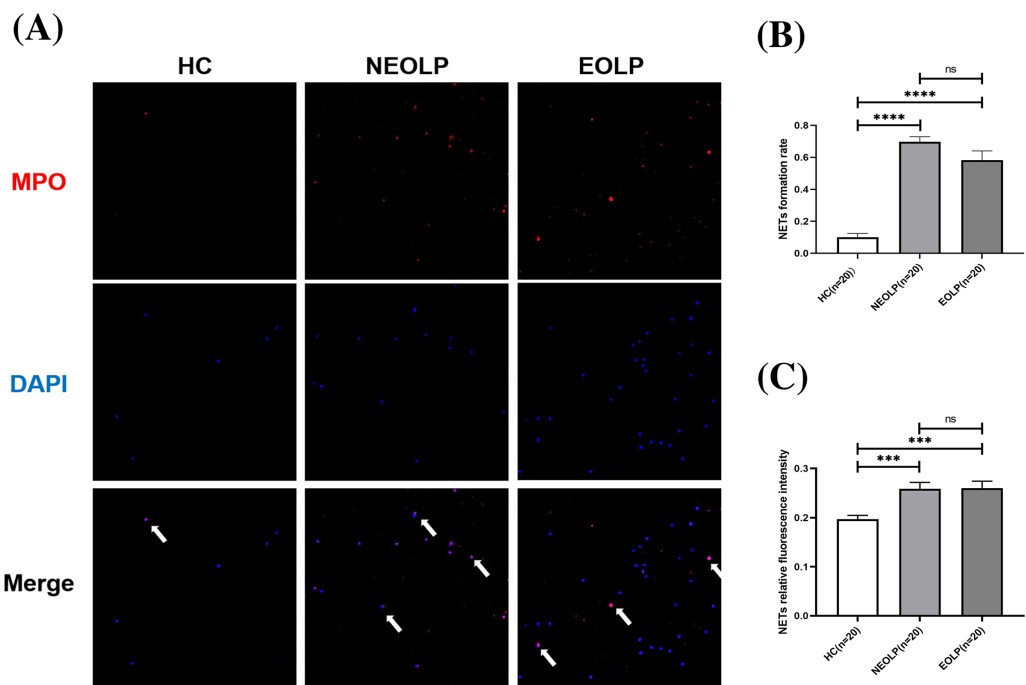

**Figure 2 (A) Expression of MPO-DNA in peripheral blood assessed by cellular immunofluorescence staining. (B) Relative rate of NET-related protein MPO-DNA formation in peripheral blood. (C) Relative fluorescence intensity of NET-related cfDNA in peripheral.** *** means $p < 0.001$, **** means $p < 0.0001$, ns means $p > 0.05$.

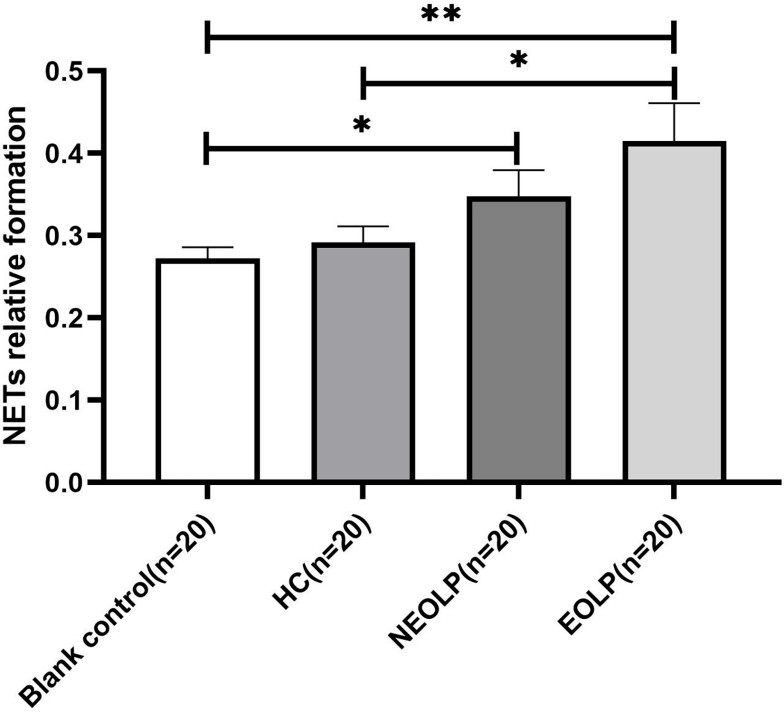

**Figure 3 Expression of NET-related cfDNA after plasma-cell coculture. NET, neutrophil extracellular trap; cfDNA, circulating free DNA.** * means $p < 0.05$, ** means $p < 0.01$.

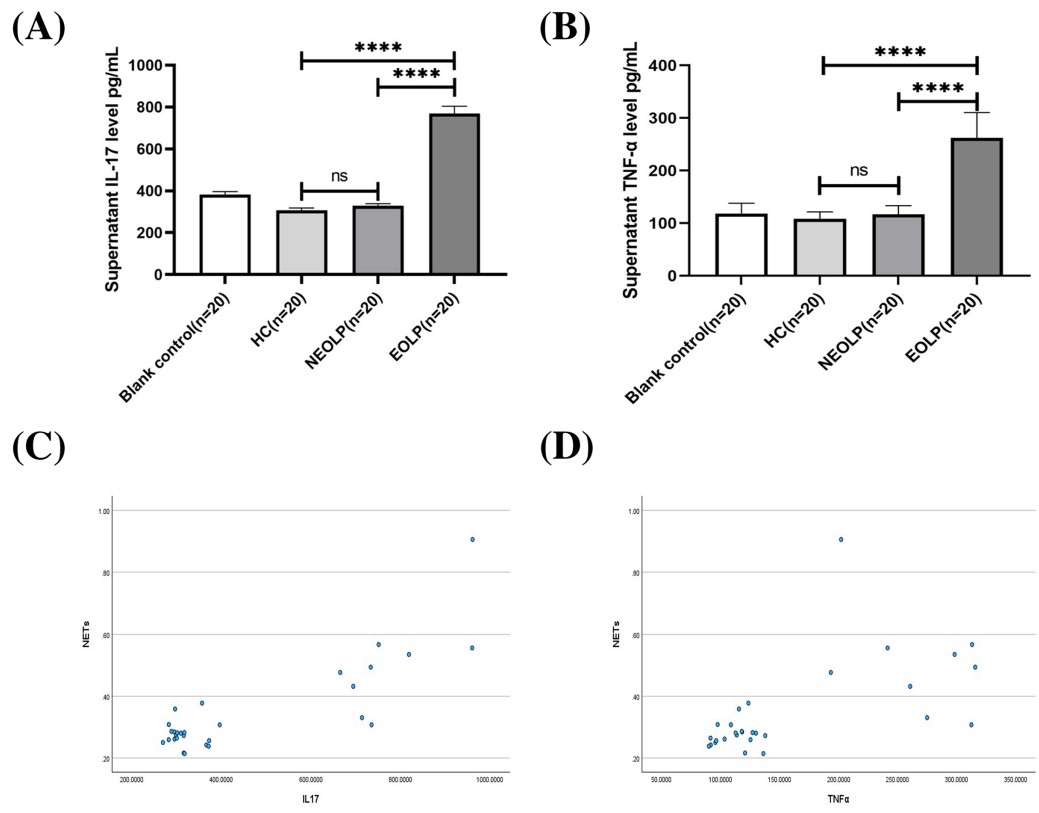

**Figure 4** Plasma levels of IL-17 (A) and TNF-α (B) in each group. Monotonicity analysis of IL-17 (C) and TNF-α (D) and its correlation with NET formation. IL-17, interleukin-17; TNF-α, tumor necrosis factor α. **** means $p < 0.0001$, ns means $p > 0.05$.

Furthermore, the results obtained from fluorescence spectrophotometry showed that the expression of NET-related cfDNA in the peripheral blood of NEOLP and EOLP patients was significantly higher than that in normal peripheral blood ($p < 0.05$). The expression of NET-related cfDNA in the peripheral blood of EOLP and NEOLP was similar, with no siginificant difference ($p > 0.05$) (Fig. 2C).

## Fluorescence spectrophotometry of NET-related cfDNA in neutrophils after plasma-cell co-culture

The results of fluorescence spectrophotometry showed that the expression of NET-related cfDNA in the EOLP group was significantly higher than that in the HC group following plasma-cell co-culture ($p < 0.05$). However, there was no significant difference observed between the NEOLP group, HC group, and EOLP group ($p > 0.05$). When compared to the blank control group, the expression of NET-related cfDNA in the NEOLP and EOLP groups was significantly higher ($p < 0.05$), while no significant difference was observed between the HC group and the blank group ($p > 0.05$) (Fig. 3).

## Expression levels of IL-17 and TNF-α in neutrophil supernatant (plasma) of each group

The ELISA results showed that the EOLP group had significantly higher levels of IL-17 and TNF-α compared to the HC and NEOLP groups, with no significant difference observed between the NEOLP and HC groups (Figs. 4A, 4B).

Monotonicity detection conducted using SPSS software showed that NETs had a positive monotonic relationship with IL-17 and TNF-α. Specifically, higher levels of IL-17 and TNF-α corresponded to increased formation of NETs (Figs. 4C, 4D). Spearman correlation analysis further confirmed these findings, with correlation coefficients of 0.568 and 0.655, respectively ($p < 0.001$), indicating a positive correlation between the formation of NETs and the levels of IL-17 and TNF-α.

## DISCUSSION

In this study, we observed an elevation in the levels of the NET-related protein MPO-DNA in OLP tissues compared to normal oral mucosal tissues. Particularly, this high expression predominantly localized within the subepithelial layer and lamina propria. Furthermore, the tissue levels of MPO-DNA were higher in EOLP than in NEOLP, suggesting a potential correlation between NETs and the inflammatory severity of OLP. In cases of EOLP, the compromised epithelial barrier facilitates the invasion of pathogens, consequently triggering a robust recruitment of neutrophils to the lesion site, thereby promoting the formation of NETs. Results of existing literature were consistent with our findings, suggesting significantly elevated expression of NET-related proteins in the lamina propria of mucosal lesions in patients with ulcerative colitis (UC), particularly in inflamed mucosa compared to non-inflamed mucosa (*Dinallo et al., 2019*). In addition, the increased expression of NET-related proteins has been observed in psoriasis skin lesions (*Herster et al., 2020*). These findings were in line with the results of our study, validating the significant upregulation of NET-related proteins in OLP. Thus, it could be inferred that neutrophils play a pivotal role in the formation of NETs within OLP lesions. The excessive production of NETs exacerbates the inflammatory response, aggravating oral mucosal tissue injury and likely contributing to the local pathological process.

In line with previous studies, our findings suggest that apart from local lesions, peripheral blood samples from OLP patients also exhibit increased expression of NETs (*Jablonska et al., 2020*). Moreover, investigations into other autoimmune diseases have revealed a similar trend, wherein increased expression of NETs in peripheral blood correlates with disease progression and inflammatory status in conditions such as SLE, RA, UC, and psoriasis (*Dinallo et al., 2019*; *Frangou et al., 2019*; *Herster et al., 2020*; *Khandpur et al., 2013*). Under various pathophysiological conditions, both *in vivo* and *in vitro* stimuli can stimulate neutrophils to form NETs (*Hidalgo et al., 2022*). Cell-free DNA (cf-DNA) is the main component of NETs. Previous studies have confirmed that rapid quantitative detection of plasma cf-DNA/NETs accurately reflects the content of cf-DNA in peripheral blood NETs, offering specificity for neutrophil-produced cf-DNA (*Mikacenic et al., 2018*; *Yokoyama et al., 2019*). In our study, peripheral blood neutrophils were detected by IF staining and cf-DNA detection. Our results indicated that the levels of NET-related

markers MPO-DNA and cf-DNA in peripheral blood neutrophils of OLP patients were increased compared to those in the HC group, consistent with the findings of previous research, indicating that the increase of NETs in peripheral blood in OLP might originate from neutrophils. The formation of NETs is not only related to the inflammation of local lesions of OLP, but also associated with the systemic conditions of OLP patients, underscoring the potential significance of NETosis in the occurrence and development of OLP.

To understand the association between NETs and OLP risk, we analyzed the expression of NET-related proteins across different subtypes of OLP. Our analysis revealed no significant correlation between their expression and OLP typing, indicating that abnormal upregulation of NET-related proteins could be detected in both NEOLP and EOLP. One explanation is that abnormal NETs formation serves as a common inflammatory risk marker, thus linking it to inflammatory diseases, including OLP. Another explanation is that the circulating markers assessed in this study reflect systemic conditions and do not directly indicate local disease typing. These two explanations collectively suggest that peripheral blood NET-related markers hold promise as potential early diagnostic biomarkers for OLP. However, further clinical studies are warranted to substantiate these findings and elucidate their clinical utility.

NETs can arise not only in response to pathogens but also in response to sterile stimuli such as IL-17 and TNF-$\alpha$ inflammatory cytokines (*Fousert, Toes & Desai, 2020*). For example, IL-17A and TNF-$\alpha$ have been demonstrated to induce NETosis in neutrophils of RA, while NETs significantly exacerbate the inflammatory response of RA synovial fibroblasts (*Khandpur et al., 2013*). TNF-$\alpha$ could stimulate neutrophils in UC to form NETs, and the expression of NETs in colon tissues of UC patients after treatment with the TNF receptor antagonist infliximab (IFX) was significantly reduced (*Dinallo et al., 2019*). However, in existing studies on OLP, the expressions of IL-17 and TNF-$\alpha$ in local tissues and peripheral blood were found to be increased (*Husein-ElAhmed & Steinhoff, 2022*; *Lu et al., 2015*). The results of plasma and cell co-culture in this study showed that neutrophils tended to spontaneously generate NETs, releasing inflammatory cytokines such as IL-17 and TNF-$\alpha$, further aggravating the inflammatory response. Compared to the blank group, the formation of NETs was significantly increased in NEOLP and EOLP groups, indicating that OLP plasma could promote the formation of NETs. Moreover, compared to the HC group, the formation of NETs in the EOLP group was significantly increased, and the level of NET formation was positively correlated with the levels of IL-17 and TNF-$\alpha$ in plasma, indicating that high levels of OLP inflammatory cytokines could promote the formation of NETs.

*Jablonska et al. (2020)* observed that the capacity of peripheral blood-isolated neutrophils to form NETs in OLP patients under stimulation was not significantly different from that in HCs, indicating that the increased formation of NETs was not attributable to specific changes in neutrophils. Rather, it suggests a greater accumulation of neutrophils and persistent stimulation of these cells by inflammatory cytokines (*Jablonska et al., 2020*). In the clinical management of OLP, it has been found that the erosive and congestive symptoms of EOLP can be better controlled with immunosuppressants,

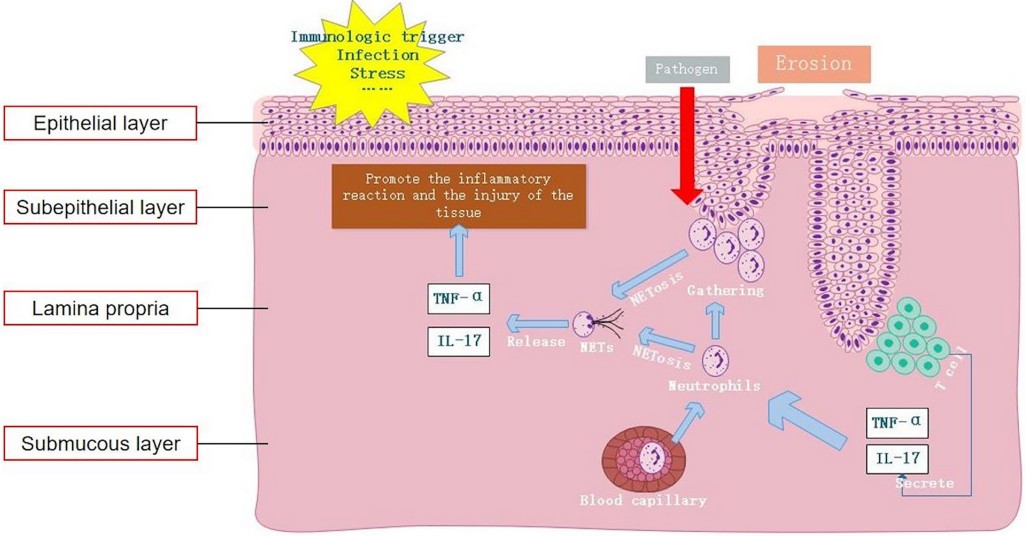

**Figure 5 Mechanism of NET formation in OLP local lesions.** Local invasive pathogens and elevated levels of inflammatory cytokines IL-17 and TNF-α recruit neutrophils to the lesion site, triggering NET formation and exacerbating the inflammatory response. NET, neutrophil extracellular trap; OLP, oral lichen planus; IL-17, interleukin-17; TNF-α, tumor necrosis factor α.

whereas the efficacy in NEOLP is limited. This indicates that the immune status of EOLP and NEOLP are different, implying distinct mechanisms for the elevated expression of NETs. Our study demonstrated that the increased level of NETs in EOLP primarily stemmed from increased formation of NETs induced by increased levels of inflammatory cytokines. In contrast, there was no significant increase in the formation of NETs in NEOLP, indicating that the high expression of NETs in NEOLP peripheral blood was unrelated to increased formation of NETs. Therefore, we speculate whether the increased level of NETs in NEOLP are associated with impaired plasma clearance function, which could also explain the phenomenon that NEOLP oral white spot lesions were not easily resolved and persisted.

Nonetheless, the study had certain limitations. Firstly, as a pilot study, it solely established a correlation between NET-related markers and OLP. Secondly, the involvement of NETs in various pathological processes suggests that its specific role in the pathogenesis of OLP warrants further investigation. In addition, the average age of the control group at the time of sample collection was lower than the average age of the patients with OLP, which may have had some influence on the results of the experiment. Patients over 50 years of age often have other, sometimes inflammatory, diseases that increase these inflammation-related markers. Subsequent studies may consider validating and extending the current results by controlling for age differences between groups, expanding sample sizes, and adding pre- and post-treatment comparisons of patients.

## CONCLUSION

In summary, this study confirmed elevated expression levels of NETs both locally within the lesions of OLP and in peripheral blood. Moreover, it demonstrated that excessive formation of NETs mediated by IL-17 and TNF-α serves as a key factor in maintaining the inflammatory response in OLP (Fig. 5). Therefore, inhibiting the formation and release of NETs and degrading the DNA backbone that constitutes NETs can be used to alleviate the harmful immune response during the disease process of OLP, serving as a potential treatment strategy. Our study provides an important clinical basis for understanding the etiology of OLP from the perspective of NET formation.

### Funding

This work was supported by the Special Fund of Jiangsu Provincial Key Research and Development Project (Social Development) (Grant No. BE2021723), the Jiangsu Province Capability Improvement Project through Science, Technology and Education-Jiangsu Provincial Research Hospital Cultivation Unit (YJXYYJSDW4), the Jiangsu Provincial Medical Innovation Center (CXZX202227), and the National Nature Science Foundation of China (Grant No. 82470979). The funders had no role in study design, data collection and analysis, decision to publish, or preparation of the manuscript.

### Grant Disclosures

The following grant information was disclosed by the authors:
Special Fund of Jiangsu Provincial Key Research and Development Project (Social Development): BE2021723.
Jiangsu Province Capability Improvement Project through Science, Technology and Education-Jiangsu Provincial Research Hospital Cultivation Unit: YJXYYJSDW4.
Jiangsu Provincial Medical Innovation Center: CXZX202227.
National Nature Science Foundation of China: 82470979.

### Competing Interests

The authors declare that they have no competing interests.

### Author Contributions

- Juehua Cheng performed the experiments, analyzed the data, prepared figures and/or tables, and approved the final draft.
- Chenyu Zhou performed the experiments, analyzed the data, prepared figures and/or tables, and approved the final draft.
- Jia Liu analyzed the data, prepared figures and/or tables, and approved the final draft.
- Yanlin Geng analyzed the data, prepared figures and/or tables, and approved the final draft.
- Lin Liu conceived and designed the experiments, authored or reviewed drafts of the article, and approved the final draft.

- Yuan Fan conceived and designed the experiments, authored or reviewed drafts of the article, and approved the final draft.

### Human Ethics

The following information was supplied relating to ethical approvals (*i.e.*, approving body and any reference numbers):

All samples were collected with the informed consent of the participants, and all procedures were approved by the Ethics Committee of the School of Stomatology, Nanjing Medical University, under approval number No. 281 (2019).

### Clinical Trial Ethics

The following information was supplied relating to ethical approvals (*i.e.*, approving body and any reference numbers):

All samples were collected with the informed consent of the participants, and all procedures were approved by the Ethics Committee of the School of Stomatology, Nanjing Medical University, under approval number No. 281 (2019).

### Data Availability

The raw measurements are available in the Supplemental File.

### Supplemental Information

Supplemental information for this article can be found online at http://dx.doi.org/10.7717/peerj.18260#supplemental-information.

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
