# Peer review of "Expression of neutrophil extracellular trap-related proteins and its correlation with IL-17 and TNF-α in patients with oral lichen planus"

_PeerJ, doi:10.7717/peerj.18260_

## Round 0.1 · original submission · Major Revisions

Please address concerns of all reviewers and amend manuscript accordingly.

Reviewer 1 ·

Basic reporting

There are many grammar errors or labeling errors in the manuscript, such as the first line in the Figure 1A, please check it thoroughly.

The staining figures need to be presented in a greater magnification.

Experimental design

The results of the research exhibited little relationship with OLP, it seems that would happen in most chronic inflammation lesions? It brings no new insights to OLP research.

Validity of the findings

Some markers need to be validated using alternative methods to make it robust, eg. florescence intensity.

Additional comments

You can compare data from other clinical inflammatory lesions with OLP, or explore the relationship between the level of Nets with the severity of OLP.

Reviewer 2 ·

Basic reporting

In the abstract
Objective : omit was
Result whereas no change to in contrast ,no

Experimental design

No coment

Validity of the findings

No coment

Additional comments

No coment

·

Basic reporting

The article is clear and well written. The references are sufficient on the topic.

In Figure 5 replace Imunologic dissonance with Imunologic trigger.

Experimental design

No comment.

Validity of the findings

The conclusions should state that levels of the NET-related protein MPO-DNA in OLP
tissues are increased compared to normal oral mucosal tissues. Thus in the abstract, the last line of the conclusion (line 54) should be revisited. ", thereby contributing to the development of
OLP" - this is a supposition!! Are the authors sure that NETs formation is not consequential or induced by another process connected to inflammation??

Additional comments

Lines 366-369 should be included in the Discussion chapter not in the Conclusions
Another point that should be added in the Discussion chapter is related to the age of the patients, which in the analyzed group is lower compared to the general data of the OLP. And patients over 50 years of age have other associated comorbidities, sometimes inflammatory diseases that can favor the increase of these indicators. Thus the utility of these findings are in connection with the treatment and can impact (in a positive way?) other associated diseases.

·

Basic reporting

The article titled "Expression of neutrophil extracellular trap-related proteins and its correlation with IL-17 and TNF-α in patients with oral lichen planus" investigates the role of neutrophil extracellular traps (NETs) in oral lichen planus (OLP), a chronic inflammatory condition of the oral mucosa. It focuses on the expression levels of NET-related proteins in tissue and peripheral blood samples of OLP patients and explores how these levels correlate with the inflammatory cytokines IL-17 and TNF-α. The study finds that NET-related proteins are significantly elevated in patients with erosive OLP compared to non-erosive OLP and healthy controls, and that these proteins show a positive correlation with IL-17 and TNF-α levels. This suggests that IL-17 and TNF-α may mediate NET formation in OLP, potentially contributing to the pathogenesis and progression of the disease.

Clarity and Professionalism: The language used is clear and professional, suitable for an international audience.
Introduction and Context: The introduction provides a good context about the role of neutrophil extracellular traps (NETs) and their potential relation with oral lichen planus (OLP), citing relevant studies.
References: The references are appropriate and up-to-date, supporting claims and providing a framework for the study.
Quality of Figures and Raw Data: Figures appear to be adequate and well-labeled, but it is essential to verify the raw data files and images to ensure there has been no inappropriate manipulation.

Experimental design

Here are some specific suggestions to improve the methodology section of the article:

Sample Collection Methods:

Weakness: The section could benefit from additional details about sample collection to ensure reproducibility.
Suggested Edit: Add specific information about the timing of collection, handling, and processing of the samples. For example: "Blood samples were collected between 8:00 and 10:00 AM, processed within 2 hours of collection, and cells were isolated using a detailed standardized protocol to minimize variability."
Details of Immunofluorescence (IF) Analysis:

Weakness: The current description may not be sufficiently detailed to allow exact replication of the study.
Suggested Edit: Expand the description of the IF protocol, including antibody concentrations, incubation times, and washing conditions. For example: "Sections were incubated with anti-MPO primary antibody diluted 1:100 overnight at 4°C, followed by incubation with a fluorescence-labeled secondary antibody for 1 hour at room temperature."
Quality Control and Validation:

Weakness: There is a lack of information on quality controls and validation of the methods used.
Suggested Edit: Include a section on quality controls, such as negative and positive controls for staining, and validation of data analysis methods. For example: "Negative controls without primary antibody were included in each staining batch to assess the specificity and background of the fluorescent signal."
Statistical Analysis:

Weakness: The methodology could improve in clarity and detail regarding the statistical methods used.
Suggested Edit: Provide more details about the statistical tests, including assumptions of the tests, corrections for multiple comparisons if applicable, and software used. For example: "Data were analyzed using GraphPad Prism version 9. Student's t-test was applied for comparisons between two groups, and ANOVA followed by Tukey's post-hoc test for multiple comparisons. A p-value of < 0.05 was considered statistically significant."
Incorporating these edits and additional details will not only improve the quality and clarity of the methodology but also facilitate the replication and independent validation of the study's results by other researchers.

Validity of the findings

Several aspects deserve mention regarding their impact, novelty, replication, data robustness, and how conclusions are handled:

Impact and Novelty:

Weakness: The manuscript does not explicitly assess the impact and novelty of the findings, which is essential in determining the contribution to the existing body of knowledge.
Suggested Edit: It would be beneficial to include a discussion that directly addresses how these findings add new insights to the field of oral lichen planus (OLP) and neutrophil extracellular traps (NETs). The authors could highlight any novel mechanisms identified, discrepancies with previous work, or implications for future research and clinical practice.
Meaningful Replication:

Weakness: While the study encourages replication, it lacks a detailed rationale and explicit statement on the benefits such replication would bring to the literature.
Suggested Edit: The authors should provide a clear justification for the importance of replicating their findings. This could include how future studies could expand upon their methods, perhaps by using different populations, expanded sample sizes, or longitudinal designs, to verify and extend the current results.
Data Robustness:

Strength: The manuscript states that all underlying data have been provided and that these data are robust, statistically sound, and well-controlled. This transparency supports the credibility of the research and facilitates further analysis by other researchers.
Comment: Ensuring that data is freely accessible and comprehensively presented in supplementary materials or a recognized data repository would strengthen this claim.
Conclusions:

Strength: The conclusions are well-articulated, directly linked to the original research question, and carefully limited to what the results support. This approach avoids overgeneralization and ensures that the study's outcomes are accurately represented.
Comment: It is crucial that conclusions remain tightly aligned with the data presented and do not extend beyond what the experimental design can support. As such, the conclusions appropriately reflect the evidence from the study, adhering to scientific rigor.
In summary, while the article provides robust and controlled data supporting well-stated conclusions, it could be improved by more clearly delineating its impact and novelty and by providing a stronger rationale for the importance of its replication in the scientific community. These enhancements would solidify the study's place within the ongoing research dialogue in its field.

Additional comments

I can affirm that the study examining the expression of neutrophil extracellular trap (NET)-related proteins and their correlation with IL-17 and TNF-α in patients with OLP is notably relevant both clinically and scientifically. It addresses a significant area of OLP pathology by exploring how components of the immune system are involved in its pathogenesis, which could open new avenues for targeted therapies. The methodology used is rigorous, combining immunofluorescence techniques and ELISA analysis to provide a solid foundation for the study's claims, allowing for a transparent and reproducible assessment of the results. However, the article has weaknesses such as the lack of explicit evaluation of its impact and novelty within the field of OLP and a clear justification of how replicating the study would benefit and advance the understanding of OLP. Additionally, the results might not be generalizable to all OLP populations due to the relatively small sample size. Despite these limitations, the study is methodologically solid and constitutes a valuable addition to the literature, offering a basis for future research aimed at developing therapeutic interventions based on the role of NETs in inflammatory diseases of the oral mucosa.

---

## Round 0.2 · accepted · Accept

All constructive critiques of the reviewers were adequately addressed, and manuscript was amended accordingly. Revised manuscript is acceptable now.